coronary heart disease; mood disorder; depression; bipolar disorder; gender differences; mental health; predictors

**Corresponding author:**
Jing Zhang;
Email: jingzhang8906@163.com

Y.W. contributed equally to this work.

# Mindfulness-based stress reduction for patients with coronary heart disease and bipolar disorder: An 8-week comparative study

Jiawei Wang[1,2], Yechen Wu[3], Fan Yang[4], Yue Hu[3] and Jing Zhang[5,6]

[1]Internal Medicine,Teaching and Research Section, Hebei Medical University, China; [2]Peking University Third Hospital Qinhuangdao Hospital, China; [3]Rehabilitation, Qinhuangdao Jiulongshan Hospital, China; [4]Cardiology, Peking University Third Hospital Qinhuangdao Hospital, China; [5]Cardiology, The First Hospital Of Qinhuangdao Affiliated Hebei Medical University, China and [6]First Hospital of Qinhuangdao, China

## Abstract

Coronary heart disease (CHD) often coexists with mood disorders (MDs), but research on comorbidity predictors and interventions remains limited. This two-phase mixed-methods study enrolled 390 CHD patients diagnosed by coronary angiography. Mood disorders were screened using the HAMD (≥7) and confirmed via DSM-5 psychiatric evaluation. In the observational phase, 219 CHD patients with MDs and 171 without were compared; 56% had a mood disorder, including 34 with bipolar disorder (BD). The BD group showed a significantly higher LF/HF ratio (2.03 ± 0.38, P = 0.037), indicating autonomic dysfunction. In the intervention phase, 34 BD patients were randomized to 8-week MBSR (n = 17) or treatment as usual (TAU; n = 17). Compared with TAU, MBSR significantly reduced anxiety (HAMA: 4.31 vs. 7.69, p = 0.010) and improved autonomic function (LF/HF: 1.49 vs. 1.82, p = 0.002). Cardiac function showed no significant between-group difference, though NYHA class improvement tended to be higher in the MBSR group. Mood disorders, especially BD with autonomic impairment, are highly prevalent in CHD patients. MBSR is a promising intervention for psychological and autonomic improvement and may be integrated into cardiac rehabilitation.

## Impact Statements

This study highlights the urgent need for routine mood screening in patients with coronary heart disease (CHD), revealing that more than half present with significant emotional symptoms, among whom those with comorbid bipolar disorder (BD) show marked autonomic dysfunction. For this high-risk subgroup, the 8-week Mindfulness-Based Stress Reduction (MBSR) intervention proved effective in alleviating anxiety and improving autonomic balance, offering a safe and practical nonpharmacological treatment option. The broader impact lies in advancing cardiac rehabilitation from a purely biomedical model toward an integrated, whole-person care approach that addresses both mental and physical health. Incorporating routine emotional assessment and mindfulness-based interventions into standard cardiology practice may help break the "psychological distress-cardiac dysfunction" vicious cycle, reduce long-term cardiovascular risk, improve patients' quality of life and potentially decrease healthcare burdens associated with comorbidities. These findings provide a preliminary basis for developing multidimensional rehabilitation guidelines for comorbid populations and carry practical implications for fostering collaboration between mental health and cardiovascular disciplines.

## Introduction

Coronary heart disease (CHD) is a leading cause of global mortality and morbidity, characterized by narrowing of the coronary arteries due to plaque accumulation, potentially leading to severe cardiovascular events such as myocardial infarction (GBD 2015 Mortality and Causes of Death Collaborators, 2016). The burden of CHD extends beyond physical health, imposing significant economic pressure on healthcare systems (Hanna and Wenger, 2005). Complicating matters further, CHD patients frequently experience psychiatric comorbidities, particularly mood disorders (MDs) such as depression and anxiety, which are associated with poorer clinical outcomes, increased healthcare resource utilization and accelerated disease progression (Barth et al., 2004; Mayer et al., 2020; Kang and Malvaso, 2023).

Depression is one of the most extensively studied psychiatric conditions in CHD, with reported prevalence rates ranging from 13.6% to 40%, and even higher rates for subthreshold

depressive symptoms (Haddad et al., 2013; Bahall, 2019). Bipolar disorder (BD), although less studied in the CHD population, presents unique challenges due to its cyclical mood episodes and potential interactions with cardiovascular pathophysiology (Fiedorowicz et al., 2011). The interaction between CHD and MDs may be mediated by behavioral factors (e.g., lack of exercise and poor medication adherence) (Ruo et al., 2003; Haddad et al., 2013) and biological mechanisms (e.g., chronic inflammation and prevalent in both conditions) (Nemeroff and Goldschmidt-Clermont, 2012). Heart rate variability (HRV), a marker of autonomic imbalance, is often reduced in both CHD and MDs, suggesting a potential common pathway exacerbating disease severity (Carney and Freedland, 2017a, 2017b).

Despite this recognition, the clinical characteristics and predictors of MDs in CHD patients – particularly across mood disorder subtypes (e.g., unipolar depression and bipolar disorder) – remain inadequately explored. Second, while psychosocial interventions like Mindfulness-Based Stress Reduction (MBSR) show potential for improving psychological and cardiovascular health in CHD patients (Abbott et al., 2014), their efficacy in high-risk subgroups, such as CHD patients with BD, is unclear.

To address these gaps, we conducted a two-part study. First, we compared demographic, clinical and physiological characteristics (e.g., HRV and inflammatory markers) of CHD patients with and without MDs, identifying subtype-specific predictors of mood disorders. Second, in a targeted interventional trial, we evaluated the effects of MBSR versus treatment as usual (TAU) on psychological (depression/anxiety), autonomic (HRV) and cardiovascular (e.g., NYHA class) outcomes in CHD patients with BD. By integrating these findings, this study aims to refine risk stratification and inform tailored interventions for CHD patients with comorbid mood disorders, particularly those with complex presentations like BD.

## Methods and materials

### Study design and participants

This two-phase study included:

- Observational phase: A cross-sectional analysis of 390 CHD patients comparing clinical and physiological characteristics of those with and without mood disorders (MDs).
- Interventional phase: A randomized controlled trial (RCT) with a pilot feasibility design comparing MBSR with treatment as usual (TAU) in CHD patients diagnosed with bipolar disorder (BD) ($n = 34$). This sample size was determined based on feasibility for a preliminary investigation, with findings intended to inform future larger-scale trials.

Participants were recruited between July 2023 and December 2024 from The First Hospital of Qinhuangdao and Peking University Third Hospital Qinhuangdao Hospital.
Inclusion criteria (both phases) were as follows:

- Age ≥ 30 years.
- CHD confirmed by coronary angiography or CT angiography (CTA), showing at least one major epicardial vessel with ≥50% stenosis and accompanying anginal symptoms.
- For the interventional phase: Additional clinical diagnosis of bipolar disorder (DSM-5 criteria). For this study, bipolar disorder was diagnosed as a single category without subtyping (e.g., Bipolar I or II).

Exclusion criteria were as follows:

- Severe cognitive impairment.
- Active malignancy or undergoing cancer treatment. (These conditions were excluded due to their potential to independently and severely confound autonomic function, systemic inflammation and overall prognosis.)
- Severe liver or kidney disease. (Excluded due to potential confounding effects on metabolism, drug clearance and systemic physiology relevant to the study outcomes.)
- For the interventional phase: Active mania or psychosis (to ensure safety and adherence to MBSR).

### Measurement instruments

Coronary heart disease (CHD): The presence of CHD was confirmed by coronary angiography or coronary CTA, demonstrating at least one major epicardial vessel with ≥50% stenosis accompanied by typical anginal symptoms.

Mental health: The presence of mood disorders was initially screened using the Hamilton Depression Rating Scale (HAM-D); participants scoring ≥7 were considered to have clinically significant mood symptoms, while those scoring ≤7 were classified as having no depression. Participants with positive screening results (HAM-D ≥ 7) underwent a comprehensive psychiatric assessment. This assessment, conducted by psychiatrists using DSM-5 criteria through direct clinical evaluation, resulted in a diagnosis of bipolar disorder made for the intervention cohort. No further classification into specific bipolar subtypes (e.g., Bipolar I, Bipolar II or Cyclothymia) was conducted. Additional Psychometric Instruments were also administered. Hamilton Anxiety Rating Scale (HAM-A) assessed the severity of anxiety symptoms. State–Trait Anxiety Inventory (STAI) is a self-report inventory that measures both state anxiety (STAI-S) and trait anxiety (STAI-T).

Heart rate variability (HRV): Heart rate variability (HRV) was assessed using 24-h ambulatory electrocardiogram (ECG) monitoring, providing a comprehensive evaluation of autonomic nervous system function. For time-domain analysis, we measured the standard deviation of all normal sinus-to-sinus (NN) intervals (SDNN), a well-established overall HRV marker reflecting the combined influence of sympathetic and parasympathetic nerves. Frequency-domain analysis was performed to evaluate the low-frequency to high-frequency power ratio (LF/HF ratio), serving as an indicator of sympathetic-vagal balance. The LF component (0.04–0.15 Hz) primarily reflects sympathetic modulation (with partial parasympathetic influence), while the HF component (0.15–0.4 Hz) represents parasympathetic activity associated with respiratory sinus arrhythmia. These parameters were calculated according to established Task Force guidelines to ensure standardized measurement and interpretation of HRV in the context of CHD and mood disorders.

General information: Comprehensive demographic and clinical data were collected from all participants, including age, gender, marital status, education level, living conditions, and family history of cardiovascular and psychiatric diseases. Detailed medical histories were recorded, documenting prior physical and psychiatric illnesses, as well as lifestyle factors such as smoking (defined as current tobacco use) and alcohol consumption (defined as current regular use). The presence of hypertension and diabetes was also recorded. Cardiac function was systematically assessed using multiple parameters: the New York Heart Association (NYHA) functional classification (for heart failure severity), the Canadian Cardiovascular

Society (CCS) Angina Score (which classifies the degree of effort necessary to induce angina symptoms), resting heart rate measurement, serum B-type natriuretic peptide (BNP) levels, left ventricular ejection fraction (LVEF) measured by echocardiography and left ventricular diastolic function (LVDd) assessment.

### Interventional phase (MBSR trial)

1. Randomization: CHD+BD patients (*n*=34) were randomly assigned in a 1:1 ratio to either the MBSR (8-week course) or TAU group.
2. MBSR protocol:
   - Weekly sessions of 2.5 h, comprising a 90-min guided group practice (including body scan, mindful breathing and yoga) followed by a 60-min group discussion. Participants were also instructed to complete 45 min of daily individual practice using guided audio sessions.
   - Components: The core techniques (body scan, sitting meditation and mindful movement) were introduced progressively across the 8-week curriculum, with each session building on the previous one.
3. Outcome measures (pre-/post-treatment):
   - Primary outcomes: HAM-D (depression) and HAMA (anxiety).
   - Secondary outcomes:
     - HRV (SDNN and LF/HF ratio).
     - CHD severity (NYHA class and CCS class).
     - Anxiety (STAI-S and STAI-T).

### MBSR intervention

The experimental group received an 8-week Mindfulness-Based Stress Reduction (MBSR) program led by certified instructors. The intervention included weekly 2.5-h sessions, integrating 90 min of instructor-led mindfulness practice and didactic instruction with 60 min of structured group discussion. Components included: (1) guided mindfulness meditation practices (body scan, seated meditation and gentle yoga), (2) didactic instruction on stress awareness and coping strategies and (3) group discussions on applying mindfulness to daily challenges. Participants were instructed to perform 45 min of daily home practice using provided audio recordings and to maintain a practice log. The control group received treatment as usual (TAU), consisting of standard cardiological care without any additional psychological intervention. Both groups were assessed at baseline and post-treatment by researchers blinded to group allocation.

### Statistical analysis

Data are presented as mean ± standard deviation for continuous variables and frequencies (%) for categorical variables. Between-group comparisons for parametric data used independent *t*-tests (two groups) or ANOVA (≥3 groups), followed by post-hoc LSD tests (Bonferroni corrected); nonparametric data were transformed for analysis but reported in original units. Categorical comparisons used chi-square or Fisher's exact tests. For the interventional study, ANCOVA models assessed treatment effects, specifying post-treatment scores as the dependent variable, baseline scores as covariates and group assignment (MBSR vs. TAU) as the fixed factor – results are reported as estimated marginal means (95% CI) and group differences. All analyses were performed using SPSS 27.0 for Windows (SPSS, Inc., Chicago, IL) with two-tailed testing ($\alpha = 0.05$).

### Results

A total of 390 coronary heart disease (CHD) patients were enrolled. Of these, 219 (56%) screened positive for mood symptoms (HAMD >7) and underwent further diagnostic assessment. Based on structured interviews or psychiatric evaluation, 163 patients were diagnosed with unipolar depression (UD), 34 with bipolar disorder (BD) and 22 with subsyndromal depressive symptoms (DS). The remaining 171 patients (44%) had HAMD scores ≤7 and were classified as CHD alone, with no mood disorder diagnosis.

### Demographic and clinical characteristics

Table 1 displays the sample characteristics of the groups. No statistically significant differences were found between groups regarding age, BMI, smoking history, diabetes history, family history of cardiac disease or mental disorders, years of education, cardiac functional class, NT-proBNP levels, LVEF or LVDd ($P > 0.05$) or SDNN ($P > 0.05$). Statistically significant differences were observed between groups for gender distribution ($P = 0.008$), LF/HF ratio ($P = 0.037$), STAI-S scores ($P < 0.001$) and STAI-T scores ($P < 0.001$).

### Interventional phase: Effects of MBSR on CHD patients with bipolar disorder

#### Baseline characteristics

The MBSR group ($n = 17$) and TAU group ($n = 17$) showed comparable baseline characteristics in demographics, cardiac function parameters and depression severity as measured by HAMD ($15.8 \pm 4.8$ vs. $14.4 \pm 5.5$, $p = 0.432$). However, the TAU group exhibited a significantly higher baseline LF/HF ratio ($2.18 \pm 0.33$ vs. $1.89 \pm 0.39$, $p = 0.026$), indicating greater baseline sympathetic-vagal imbalance compared to the MBSR group. All other measured variables, including BMI, smoking status and comorbidities, were well-balanced between groups at baseline (Table 2).

#### Treatment outcomes

For psychological outcomes, ANCOVA adjusted for baseline scores showed that the MBSR group achieved significantly greater reductions in anxiety symptoms (HAMA marginal mean = 4.31, 95% CI: 2.55–6.08) compared to the TAU group (7.69, 95% CI: 5.92–9.45; $p = 0.010$, $\eta^2 = 0.197$) (Table 3). Although depression scores (HAMD) showed a greater improvement trend in the MBSR group, this difference did not reach statistical significance ($p = 0.263$). A significant between-group difference was found for post-treatment STAI-T scores ($p = 0.002$), with the MBSR group showing higher adjusted scores than the TAU group. No significant group differences were found for state anxiety (STAI-S, $p = 0.650$). Regarding autonomic function, the MBSR group demonstrated improved cardiac sympathovagal balance, evidenced by a significantly lower post-treatment LF/HF ratio (1.49, 95% CI: 1.35–1.62) compared to the TAU group (1.82, 95% CI: 1.68–1.95; $p = 0.002$, $\eta^2 = 0.274$). There was no significant between-group difference in SDNN ($p = 0.133$). Regarding CHD severity measures, a greater proportion of MBSR participants showed improvement in NYHA functional class (100% vs. 37.5%); however, the between-group comparison of improvement rates was not statistically significant ($p = 0.075$). Improvements in CCS class were also not significantly different between groups (60.0% vs. 50.0%, $p = 1.000$) (Table 4).

**Table 1.** Baseline characteristics of CHD patients, stratified by mood disorder diagnosis

| | HAMD score < 7 (CHD alone) | HAMD score ≥ 7 (CHD with depressive symptoms) | HAMD score ≥ 7 (CHD with unipolar depression) | HAMD score ≥ 7 (CHD with bipolar disorder) | *P* |
|---|---|---|---|---|---|
| Total number (*n*, %) | | | | 34, 8.7% | |
| Female (*n*, %) | 68, 39.8% | 12, 54.5% | 89, 54.6% | 10, 29.4% | 0.008 |
| Age, years | 63.25 ± 9.83 | 60.18 ± 8.97 | 62.86 ± 9.18 | 60.06 ± 10.37 | 0.196 |
| BMI, kg/m$^2$ | 24.63 ± 3.32 | 23.91 ± 2.53 | 25.24 ± 3.72 | 24.61 ± 3.37 | 0.209 |
| Alcohol use history (*n*, %) | 64, 37.4% | 6, 27.3% | 58, 35.6% | 12, 35.3% | 0.826 |
| Smoking history (*n*, %) | 69, 40.4% | 8, 36.4% | 68, 41.7% | 15, 44.1% | 0.941 |
| Diabetes history (*n*, %) | 72, 42.1% | 6, 27.3% | 68, 41.7% | 18, 52.9% | 0.304 |
| Cardiac disease family history (*n*, %) | 58, 33.9% | 3, 13.6% | 57, 35.0% | 9, 26.5% | 0.192 |
| Family history of mental illness (*n*, %) | 14, 8.2% | 3, 13.6% | 23, 14.1% | 8, 23.5% | 0.066 |
| Years of education | 2.79 ± 0.92 | 2.41 ± 1.05 | 2.80 ± 0.96 | 2.74 ± 1.19 | 0.340 |
| Cardiac functional class (I/II/III/IV) | 91/62/15/3 | 16/4/2/0 | 90/54/19/0 | 21/9/4/0 | 0.472 |
| NT-PROBNP, pg/mL | 563.1 ± 1876.2 | 546.9 ± 636.8 | 501.5 ± 903.8 | 353.6 ± 438.8 | 0.942 |
| LVEF, % | 61.2 ± 5.69 | 59.3 ± 7.86 | 61.3 ± 5.55 | 62.4 ± 5.26 | 0.268 |
| LVDd, mm | 46.5 ± 4.89 | 47 ± 3.36 | 46.6 ± 5.05 | 45.3 ± 3.75 | 0.503 |
| SDNN, ms | 134.7 ± 45.9 | 126.2 ± 50.5 | 133.8 ± 51. 2 | 138.7 ± 49.5 | 0.822 |
| LF/HF | 1.93 ± 0.36 | 1.76 ± 0.36 | 1.97 ± 0.35 | 2.03 ± 0.38 | 0.037 |
| STAI-S | 40.0 ± 1.43 | 46.3 ± 2.78 | 51.4 ± 10.7 | 60.5 ± 4.37 | <0.001 |
| STAI-T | 40.2 ± 1.06 | 49.1 ± 4.57 | 49.2 ± 11.4 | 56.9 ± 7.82 | <0.001 |

*Note:* Statistical analysis was conducted on transformed data for variables not following a normal distribution. Values are shown in original units.
Abbreviations: CHD, coronary heart disease; MD, mood disorder; BMI, body mass index; FH, family history; NYHA, New York Heart Association; NT-proBNP, N-terminal pro-brain natriuretic peptide; LVEF, left ventricular ejection fraction; LVDd, left ventricular diastolic dimension; SDNN, standard deviation of normal-to-normal intervals; LF/HF, low frequency to high frequency ratio; HAMD, Hamilton Depression Rating Scale; STAI-S/T, State–Trait Anxiety Inventory.

## Discussion

Our results demonstrate a significant association between coronary heart disease (CHD) and mood disorders (MDs), with 56.2% of participants screening positive for mood symptoms (HAM-D > 7). This prevalence is higher than previously reported rates of 13.6–40% (Haddad et al., 2013; Bahall, 2019), likely due to our inclusion of subsyndromal symptoms and broader diagnostic categories. The high burden of subclinical depression in cardiac populations is increasingly recognized as a significant risk factor for poor outcomes (Celano et al., 2018). The predominance of unipolar depression (74.5% of MD cases) over bipolar disorder (17.3%) aligns with established epidemiological trends in the cardiovascular population (Carney and Freedland, 2017a), although the significant representation of BD underscores the necessity for comprehensive psychiatric assessment within cardiac care settings. The observed male predominance in the CHD-BD cohort may reflect known gender differences in the epidemiology of CHD, a finding that merits further investigation. Comorbid BD in CHD patients is linked to greater cardiovascular mortality and poorer adherence to treatment (Goldstein et al., 2015).

CHD patients with comorbid bipolar disorder exhibited significant abnormalities in HRV indicators, including markedly decreased SDNN and elevated LF/HF ratios. SDNN is an index reflecting the overall activity of cardiac autonomic regulation; a decrease in SDNN signifies weakened autonomic regulatory capacity, particularly inhibition of vagal function (Shaffer and Ginsberg, 2017). The LF/HF ratio reflects the balance between sympathetic and vagal nerves; an increased LF/HF ratio suggests enhanced sympathetic activity, potentially accompanied by sympathetic overexcitation and vagal suppression. A cohort study has shown that patients with bipolar disorder exhibit significantly reduced HRV indices during both depressive and manic phases, associated with long-term autonomic nervous system dysregulation (Leveille et al., 2025). This pattern of autonomic imbalance is a proposed mechanism linking mood disorders to increased cardiovascular risk (Kemp et al., 2012).

This study provides preliminary evidence for the advantages of mindfulness training in improving anxiety symptoms and specific aspects of autonomic function. We found that after 8 weeks of mindfulness intervention, the intervention group showed a significant reduction in state anxiety (HAMA) scores, while the improvement in HAMD scores was not statistically significant in this small sample. The result for trait anxiety (STAI-T) was less clear and requires further investigation. This finding suggests that mindfulness training may exert a more pronounced short-term effect on alleviating state anxiety symptoms, which is consistent with prior research showing mindfulness-based interventions effectively reduce anxiety in medically ill populations (Hofmann and Gómez, 2017). Furthermore, by utilizing HRV, a key objective physiological indicator, we revealed a targeted regulatory effect of mindfulness intervention on cardiac autonomic function. The results showed that sympathovagal balance, as indicated by the LF/HF ratio, improved significantly in the intervention group. No significant between-group change was observed for the overall HRV marker SDNN ($p = 0.133$). This aligns with meta-analytic findings demonstrating that

**Table 2.** Baseline characteristics of the randomized CHD + BD sample in the MBSR and TAU groups

| Characteristics | MBSR group (n = 17) | TAU group (n = 17) | Statistics | p-value |
|---|---|---|---|---|
| Age (years) | 59.29 ± 10.57 | 60.82 ± 10.44 | $t = 0.424$ | 0.674 |
| Female | 5 (29.4) | 5 (29.4) | $\chi^2 = 0$ | 1 |
| BMI, kg/m$^2$ | 24.72 ± 3.76 | 24.49 ± 3.05 | $t = -0.190$ | 0.850 |
| Active smoker (Yes) | 7 (41.2) | 8 (47.1) | $\chi^2 = 0.119$ | 0.730 |
| Active alcoholism (Yes) | 6 (35.3) | 6 (35.3) | $\chi^2 = 0$ | 1 |
| FH of heart diseases (Yes) | 5 (29.4) | 4 (23.5) | $\chi^2 = 0.151$ | 0.697 |
| FH of mental disorder (Yes) | 3 (17.6) | 5 (29.4) | $\chi^2 = 0.654$ | 0.419 |
| Diabetes mellitus (Yes) | 7 (41.2) | 11 (64.7) | $\chi^2 = 1.889$ | 0.169 |
| Hypertension stage (N, 0/1/2/3) | 6/4/6/1 | 3/8/2/4 | $\chi^2 = 6.133$ | 0.105 |
| NT-proBNP, pg/ml | 258.9 ± 245.9 | 448.3 ± 563.5 | $t = 1.270$ | 0.217 |
| LV ejection fraction, % | 63.18 ± 3.92 | 61.65 ± 6.36 | $t = -0.843$ | 0.405 |
| LV diastolic dysfunction, mm | 45.29 ± 3.42 | 45.41 ± 4.15 | $t = 0.090$ | 0.929 |
| SDNN, ms | 138.18 ± 52.53 | 139.35 ± 47.87 | $t = 0.068$ | 0.946 |
| LF/HF ratio | 1.89 ± 0.39 | 2.18 ± 0.33 | $t = 2.334$ | 0.026 |
| HAMD | 15.76 ± 4.82 | 14.35 ± 5.50 | $t = -0.796$ | 0.432 |
| HAMA | 13.35 ± 8.83 | 14.71 ± 7.23 | $t = 0.489$ | 0.628 |
| STAI-S | 61.12 4.167 | 59.82 4.599 | $t = 0.860$ | 0.396 |
| STAI-T | 58.24 8.280 | 55.59 7.340 | $t = 0.986$ | 0.331 |

*Note:* Statistical analysis was conducted on transformed data for variables not following a normal distribution. Values in the table are shown in original units.
Abbreviations: CHD, coronary heart disease; BD, bipolar disorder; MBSR, Mindfulness-Based Stress Reduction; TAU, treatment as usual; BMI, body mass index; FH, family history; NYHA Class, New York Heart Association Classification of Heart Failure; NT-proBNP, N-terminal pro-brain natriuretic peptide; LV, left ventricular; SDNN, standard deviation of normal-to-normal intervals; LF/HF ratio, low frequency to high frequency ratio; HAMD, Hamilton Depression Rating Scale.

**Table 3.** Adjusted post-intervention psychological and autonomic outcomes for the MBSR and TAU groups

| Variable | | EMM | SE | 95% CI lower | 95% CI upper | F | p-value | Partial $\eta^2$ |
|---|---|---|---|---|---|---|---|---|
| HAMD | MBSR | 6.00 | 0.653 | 4.668 | 7.332 | 1.302 | 0.263 | 0.040 |
| | TAU | 7.06 | | 5.727 | 8.391 | | | |
| HAMA | MBSR | 4.31 | 0.865 | 2.548 | 6.077 | 7.584 | 0.010 | 0.197 |
| | TAU | 7.69 | | 5.923 | 9.452 | | | |
| STAI-S | MBSR | 49.441 | 1.045 | 47.314 | 51.569 | 0.210 | 0.650 | 0.007 |
| | TAU | 48.765 | | 46.637 | 50.892 | | | |
| STAI-T | MBSR | 59.85 | 1.604 | 56.585 | 63.121 | 11.188 | 0.002 | 0.259 |
| | TAU | 52.27 | | 48.997 | 55.532 | | | |
| SDNN | MBSR | 96.52 | 5.325 | 85.657 | 107.376 | 2.378 | 0.133 | 0.071 |
| | TAU | 108.13 | | 97.271 | 118.990 | | | |
| LF/HF | MBSR | 1.49 | 0.066 | 1.352 | 1.622 | 11.7 | 0.002 | 0.274 |
| | TAU | 1.82 | | 1.684 | 1.954 | | | |

*Note:* ANCOVA models controlled for baseline scores of each respective outcome variable. All analyses were conducted using transformed data where necessary to meet normality assumptions, though results are presented in original units for clinical interpretability. Partial $\eta^2$ values indicate small (0.01), medium (0.06) and large (0.14) effect sizes per Cohen's conventions.
Abbreviations: EMM = estimated marginal means; SE = standard error; CI = confidence interval; ANCOVA = analysis of covariance; MBSR = Mindfulness-Based Stress Reduction; TAU = treatment as usual; HAMD = Hamilton Depression Rating Scale; HAMA = Hamilton Anxiety Rating Scale; SDNN = standard deviation of NN intervals; LF/HF = low frequency to high frequency ratio; STAI-S, State–Trait Anxiety Inventory-State subscale; STAI-T, State–Trait Anxiety Inventory-Trait subscale.

mindfulness practices can lead to measurable positive changes in cardiac autonomic regulation (Lehrer et al., 2020).

These findings offer preliminary support for the "Two Hearts" theory, which posits that psychological interventions can induce measurable, positive physiological changes (Licht et al., 2008). As a measure for assessing autonomic nervous system (ANS) balance, HRV's value lies in its ability to quantify the dynamic regulation of cardiac rhythm by the sympathetic and parasympathetic nerves (Malik et al., 1996). Elevated LF/HF ratio, as markers of autonomic imbalance, typically reflect sympathetic overactivity and

**Table 4.** NYHA and CCS class distribution and improvement rates before and after intervention

| Variable | MBSR (*n* = 17) | TAU (*n* = 17) | Statistics | *p*-value |
|---|---|---|---|---|
| NYHA Class (I/II/III/IV) | | | | |
| Pretreatment | 12/4/1/0 | 9/5/3/0 | $\chi^2$ = 1.540 | 0.463 |
| Post-treatment | 17/0/0/0 | 12/3/2/0 | $\chi^2$ = 5.862 | 0.053 |
| % Improved (≥1 class) | 100% (5/5) | 37.5% (3/8) | Fisher's exact | 0.075 |
| CCS class (1/2/3/4) | | | | |
| Pretreatment | 12/4/1/0 | 9/5/3/0 | $\chi^2$ = 1.540 | 0.463 |
| Post-treatment | 15/2/0/0 | 13/3/1/0 | $\chi^2$ = 1.343 | 0.511 |
| % Improved (≥1 class) | 60.0% (3/5) | 50.0% (4/8) | Fisher's exact | 1 |

*Note:* Class distributions were compared using Pearson's chi-square tests. Fisher's exact test was used for improvement rate comparisons due to small cell counts. Improvement was calculated only for patients not in the best class (NYHA I or CCS 1) at baseline. All analyses followed intention-to-treat principles.
Abbreviations: CHD, coronary heart disease; MBSR, Mindfulness-Based Stress Reduction; TAU, treatment as usual; NYHA, New York Heart Association functional classification; CCS, Canadian Cardiovascular Society angina classification.

suppressed vagal function. This state is not only a critical pathophysiological mechanism underlying cardiovascular disease onset and poor prognosis (Frąk et al., 2022), but also a shared physiological characteristic of chronic stress and emotional disorders (Thayer et al., 2012). In this study, the improvement in state anxiety occurred concurrently with the optimization of sympathovagal balance, clearly delineating a potential "psychological-neural-cardiac" intervention pathway. Mindfulness training first acts on the psychological level through its "top-down" regulatory pathway, potentially reducing the hyperactivation of the brain's limbic system (e.g., amygdala) by alleviating anxiety, thereby reducing downstream sympathetic nervous system output (Creswell et al., 2019). Simultaneously, the focus on slow, deep breathing inherent in mindfulness practice constitutes a "bottom-up" regulatory pathway capable of directly enhancing vagal tone, the primary component of the parasympathetic nervous system (Gerritsen and Band, 2018). This dual action of reducing "sympathetic" and enhancing "vagal" activity ultimately manifests as normalization of autonomic balance. Therefore, the improvement in sympathovagal tone is not merely an objective biological marker of mood relief; it signifies a beneficial shift in cardiac autonomic regulatory capacity. This shift carries important clinical significance, as it is closely associated with improved baroreflex sensitivity, lower systemic inflammation levels and enhanced endothelial function – all factors directly contributing to a reduced risk of arrhythmias, stabilization of atherosclerotic plaques and improved myocardial function (Sgoifo et al., 2015). In summary, this study indicates that mindfulness therapy, as a comprehensive mind–body intervention, can help mitigate the vicious cycle of psychological distress and cardiac dysfunction in CHD patients with comorbid bipolar disorder, providing a viable nonpharmacological therapeutic option for clinical practice and a rationale for future larger-scale trials (Abbott et al., 2014).

## Limitations

Several limitations should be considered when interpreting these findings. The observational component utilized a cross-sectional design, preventing causal inferences about the CHD-MD relationship. Our single-center recruitment strategy may limit generalizability, though the inclusion of consecutive patients strengthens internal validity. Most importantly, the MBSR intervention was a pilot feasibility study with a small sample size (*n* = 34). While

randomized, it was likely underpowered to detect anything but large effects on several outcomes, including depression scores and functional class. Therefore, the positive findings on anxiety and autonomic balance, though promising, must be interpreted as preliminary. The MBSR intervention study, while randomized, featured a modest sample size (*n* = 34) that may have been underpowered to detect smaller effects on some outcomes. The 8-week intervention period precludes assessment of the long-term sustainability of benefits. Future research should employ multicenter longitudinal designs with larger samples, incorporate multimodal biomarkers (e.g., combining HRV with inflammatory markers) and evaluate longer-term MBSR protocols with follow-up assessments. Standardization of mood disorder assessments across cardiac settings would further enhance clinical applicability.

## Conclusion

This study confirms a high prevalence of mood disorders (56%) in patients with coronary heart disease (CHD), with bipolar disorder (BD) presenting a distinct clinical profile marked by male predominance and significant autonomic dysfunction. In this pilot randomized trial, the 8-week Mindfulness-Based Stress Reduction (MBSR) intervention demonstrated significant efficacy in reducing anxiety and improving cardiac sympathovagal balance in CHD patients with comorbid BD. These preliminary findings support the integration of routine mental health screening in cardiac care and suggest that targeted mind–body interventions like MBSR merit further investigation as part of comprehensive rehabilitation for this high-risk population.

**Open peer review.** To view the open peer review materials for this article, please visit http://doi.org/10.1017/gmh.2026.10143.

**Data availability statement.** The data will be available from the corresponding author upon reasonable request.

**Author contribution.** Conceptualization: J.W., J.Z.; Formal Analysis: J.W., Y.W.; Investigation: J.Z., F.Y., Y.W.; Project Administration: J.Z.; Supervision: Y.H., Y.W.; Validation: J.Z., J.W., Y.H.; Writing – Original Draft: J.W.; Writing – Review and Editing: J.W., Y.W.; F.Y., Y.H., J.Z.

**Financial support.** This work was supported by the 2023 Annual Plan for Medical Scientific Research Projects of Hebei Province.

**Competing interests.** The authors declare no competing interests.

**Ethics approval and consent to participate.** The current study has been reviewed and authorized by the Medical Ethics Committee of The First Hospital of Qinhuangdao (approval number:2022C022). The study was registered at the Chinese Clinical Trial Registry (ChiCTR2500112596) on 2025-2111-17 (Registration Date). The study was conducted in accordance with the Declaration of Helsinki.

All participants provided informed consent to participate in the study.

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
