## [Reviewer Report]

Reviewer comments:

A good pilot study paper on BD and CHD especially with increasing trends of burden and mortality from CVD and mental illness.

1. The sample size of 34 for 2 arms

randomisation for a quantitative study is rather small especially and not seems to be justified by

any statistically backing or minimum sample

accepted for ( quasi) experimental design of 30

per arm.( Brown ,1995). Therefore the conclusion

wording should be guarded/ revised., to

acknowledge the positive stats and thus drive for

a larger sample.

2. There are additional tools/ measures that are not mentioned in methodology nor in intervention

procedure yet appear in results table 3&4

analysis... CCS, STAI-S, STAI-T,

3. Page 7 line 46.. intervention is reported as 2.5

hours weekly sessions and in page 8 under

description of intervention, there is 90 minutes

group session with 30 minutes daily individual

audio sessions. Please clarify / harmonize on

the intervention duration. Also did all the

sessions include all the various techniques or

was it staggered in particular sessions?

4. Page 23... Add the abbreviation for HAMA

5. Discussion on the findings is good though

interpretation / conclusion and therefore

recommendations may be overly limited given

the sample size.

7. In previous literature is there any explanation on why the male predominance of BD than females among CHD patients?

significant

---

## [Reviewer Report]

This manuscript combines an observational study with a randomized controlled trial to first compare the clinical and physiological characteristics of patients with coronary heart disease (CHD) with and without mood disorders and then to assess the impact of an 8-week mindfulness-based stress reduction (MBSR) program on depression and anxiety as well as autonomic functioning in a cohort of patients with bipolar disorder (BD) and CHD. A more complete understanding of the relationships between mood disorders and coronary heart disease is important in the field despite many prior studies evaluating similar questions. Where this study shines is in attempting to evaluate the effects of MBSR in a population of patients with both BD and CHD, and it is the first study to do so to my knowledge. Unfortunately, the data presented in the manuscript are inconsistent, potentially inaccurate, and difficult to interpret and do not always seem to support the described results and conclusions in the text. As a result the study is not publishable in its current form.

Major Points

1) The methods do not sufficiently describe the diagnostic categories for the mental health conditions. Since they are diagnosed from DSM-5, depressive disorders should be classified more specifically, e.g. major depressive disorder, persistent depressive disorder, etc. Bipolar disorder should be specified as bipolar type 1, type 2, or other. Frequency of comorbid psychiatric disorders would also help the reader understand the population of patients being studied.

2) Rating scales for mental health are not sufficiently described in the methods. The State and Trait Anxiety Inventory is never mentioned despite being reported in the results, and the HAM-A is not described in the measurement instruments section despite being a primary outcome variable.

3) Definitions for smoking status and alcohol consumption are not provided, and it is unclear how alcohol/smoking history in Table 1 were differentiated from “Active Smoker” and “Active Alcoholism” in Table 2.

4) The total numbers of patients reported in the “Cardiac Functional Class” row in Table 1 do not add up to the total number of patients in the group except in the bipolar disorder cohort. Furthermore, percentage calculation for “Female” appears to be based on the 154 patients reported in the CFC row rather than the 171 patients total.

5) Baseline SDNN for the bipolar group is reported to be 115.15 ms, but in the intervention groups it is subsequently reported as 138.18 and 139.35 ms. Then in the MBSR and TAU groups post-intervention, it is 96.52 and 108.13 ms respectively. Discussion on page 17, line 41 says that the intervention increased SDNN, but this does not seem to be the case based on the presented data, nor was the comparison significant.

6) STAI-T is described as having a significantly greater reduction in the MBSR group, but in Table 3 the MBSR group has a higher STAI-T score than the TAU group, and the STAI-T score is higher even than the baseline measurement for the bipolar cohort.

7) Percentage improvements in NYHA and CCS class scores reported in Table 4 are confusing. It is not clear how 5/17 is 62.5%. Also, the text on page 13, line 55 reports the p-value from the chi-squared comparison of groups rather than the Fisher exact test for the improvement rate comparison. In either case, the results should have been non-significant.

8) Throughout the manuscript, nonsignificant results are reported as trends. I favor simply reporting them as nonsignificant differences.

Minor Points

1) Why were active malignancy and severe liver or kidney disease exclusion criteria?

2) The p-value comparisons in Table 1 are confusing. There are seemingly multiple comparisons being reported, an initial overall comparison of the distributions between all 4 groups of CHD patients (or is it just CHD patients without mood disorders vs CHD patients with mood disorders?) and then comparisons within each cell with the corresponding group (e.g. female vs male). Are the second comparisons relevant/necessary to report?

3) HAM-A baseline scores should be reported in Table 1.

4) Table 1 is missing descriptions of abbreviations used.

5) Title of Table 3 should be more descriptive of what the results are rather than “Output of ANCOVA”

---

## [Editor Report]

This study address an important clinical gap, however, it has major methodological and data-integrity problems. Diagnostic definitions are incomplete, several instruments used in the results are missing from the methods, Key tables contain numerical inconsistencies and lack clear, well-described titles, and some findings are inaccurately interpreted. The two reviewers identified substantial revisions needed to ensure accuracy, transparency, and coherence before it could be considered for publication. Given the extent of correction required, I recommend major revision.

---

## [Editor Report]

I congratulate the authors. They have responded comprehensively and satisfactorily to the editorial, my suggestion and reviewer comments, improving methodological clarity, data accuracy, and interpretative rigor. I therefore recommend the manuscript for publication, with just only one minor comment I have noted below.

- The manuscript states “Adjusted Post-Intervention Psychological and Autonomic Outcomes for the MBSR and TAU Groups” but the rationale and process for covariate selection is not well. Authors can list all covariates (e., g, age, sex etc) that were included in each adjusted model and alsodescribe model assumptions and diagnostics.